# Chemical control of the viscoelastic properties of vinylogous urethane vitrimers

Wim Denissen[1], Martijn Droesbeke[1], Renaud Nicolaÿ[2], Ludwik Leibler[2], Johan M. Winne[1] & Filip E. Du Prez[1]

Vinylogous urethane based vitrimers are polymer networks that have the intrinsic property to undergo network rearrangements, stress relaxation and viscoelastic flow, mediated by rapid addition/elimination reactions of free chain end amines. Here we show that the covalent exchange kinetics significantly can be influenced by combination with various simple additives. As anticipated, the exchange reactions on network level can be further accelerated using either Brønsted or Lewis acid additives. Remarkably, however, a strong inhibitory effect is observed when a base is added to the polymer matrix. These effects have been mechanistically rationalized, guided by low-molecular weight kinetic model experiments. Thus, vitrimer elastomer materials can be rationally designed to display a wide range of viscoelastic properties.

[1] Department of Organic and Macromolecular Chemistry, Polymer Chemistry Research Group and Laboratory for Organic Synthesis, Ghent University, Krijgslaan 281 S4-bis, Ghent B-9000, Belgium. [2] Matière Molle et Chimie, UMR 7167 CNRS-ESPCI, ESPCI ParisTech, 10 rue Vauquelin, Paris 75005, France. Correspondence and requests for materials should be addressed to J.M.W. (email: johan.winne@ugent.be) or to F.E.D.P. (email: filip.duprez@ugent.be).

Thermosets and elastomers offer substantial advantages in terms of mechanical properties and solvent resistance compared to thermoplastics, due to their cross-linked structure. However, their permanent and rigid molecular architecture is also their main drawback, as thermosets are static materials that cannot undergo any processing after full curing. Recent approaches aim to transform these permanent polymeric networks into dynamic systems using cross-links based on intermolecular interactions[1,2], reversible covalent (and/)or dynamic covalent bonds[3–5] to enable processing, recycling and even self-healing. Intermolecular interactions, together with reversible covalent bonds, rely on a triggered displacement of the association equilibrium towards the endothermic dissociated state, transforming the polymeric network into a thermoplastic or oligomer melt, depending on the network topology. This equilibrium displacement invariably gives rise to a sudden drop in viscosity of several orders of magnitude over a small temperature interval. While this sharp viscosity drop enables fast processing, it can also result in uncontrolled deformation and thus requires a precise control of the processing temperature. Moreover, the gel- to sol-transition, resulting from the cross-link density decrease, makes these reversible networks prone to the influence of solvents at high-temperatures. As an alternative to covalent networks relying on reversible dissociation of covalent bonds, Leibler and co-workers introduced the concept of vitrimers[6–8], that is, malleable networks based on thermally triggered associative exchange reactions. As network bonds are only broken when new bonds are formed, for example, through an addition/elimination mechanism, vitrimers are characterized by a constant cross-link density and thus remain insoluble at all times and any given temperatures. In addition, the viscosity decrease of vitrimers is gradual (Arrhenius behaviour) in comparison to thermoplastic melts because the flow is mainly controlled through chemical reaction rates, rather than by chain friction as for usual polymer melts. As a consequence, vitrimers are processable over a broad temperature range without the need of moulds to prevent loss of structural integrity.

Currently, as recently reviewed by our group[9], only a limited number of associative exchange chemistries[10–24] have been explored in the context of vitrimers, and one of the main challenges is the precise control of the exchange kinetics. Although Leibler and co-workers demonstrated that epoxy-based transesterification vitrimers can be controlled by changing catalysts[8], the exchange reaction remains slow and high catalyst loadings and temperatures are required to enable processing in a reasonable timeframe[6–8]. Other systems rely on exchange reactions that are already fast at room temperature. This room temperature dynamic behaviour can lead to interesting self-healing properties[18,25], but the low-temperature exchange reactions can also give unwanted deformation at service temperature for low-glass transition temperature ($T_g$) materials. Thus, vitrimer elastomers are very challenging materials as a precise control of the kinetics of exchange is required to enable preferentially fast processing at elevated temperatures (high-rate of exchange) and dimensional stability, that is, no creep at service temperature (no or very low-rate of exchange). In this work, we aimed to use simple additives that enable the precise control of exchange kinetics and subsequent viscoelastic properties of vitrimers.

Previously, we reported vitrimers based on the amine exchange of vinylogous urethanes[17], an exchange reaction that does not require a catalyst and is fast at temperatures above 100 °C, but has a negligible exchange rate at room temperature. Herein, we first expand the possibilities of dynamic vinylogous urethane chemistry platform by showing that a precise control of the exchange kinetics can be achieved using a variety of simple

additives. This precise control of the exchange reaction allows for the rational design of elastomers that can be processed with short relaxation times, and show only little creep at service temperature. Furthermore, we demonstrate that a fast exchange reaction can also be decelerated to achieve dimensional stability of elastomer vitrimers. Finally, high $T_g$ thermoset materials with an extremely fast acid-catalysed stress–relaxation have also been prepared, which thus allows for an increased processing ability.

## Results

**Model compound study**. To confirm our expectation that the amine exchange of VU, obtained from the condensation reaction between acetoacetate and amine moieties, can be controlled by additives and to gain further insight into the factors that govern the exchange dynamics, we first prepared and investigated low-molecular weight compounds in a model study. The exchange experiments were designed in a way that mimics the VU polymer matrix as closely as possible, that is, a vinylogous urethane functional group in each repeating unit and only a few reactive free amines as network defects. Thus, no solvent was used and a five-fold excess of VUs versus amines was employed in the presence of small quantities of additives (Fig. 1a). We surmised that proton transfers are essential steps during the exchange process and that protonated species are important reaction intermediates. Therefore, the effect of acids and bases was tested on the exchange kinetics of two simple model compounds. While $p$-toluene sulphonic acid ($p$TsOH) and sulphuric acid ($H_2SO_4$) were selected as acids, triazabicyclodecene (TBD) was used as strong base. These additives effectively control the amount of protonated (ammonium-type) species present in the reaction medium. In addition, dibutyltin dilaurate (DBTL) was examined as it is a widely used Lewis acid catalyst. Furthermore, these additives were chosen to exhibit a boiling point far above processing temperature in order to avoid evaporative loss. Simple carboxylic acids were considered less useful because they could irreversibly react with amines when heated. The reaction of $N$-octyl vinylogous urethane model compound with 2-ethyl hexyl amine (2-EHA) was performed at 100 °C under inert atmosphere and monitored by the disappearance of 2-EHA and appearance of n-octyl amine using gas chromatography (GC) with a flame ionization detector (FID) (Supplementary Fig. 1). The control experiment (without additives) showed that more than one hour of heating is needed to reach equilibrium, which corresponds to a remaining fraction of 2-EHA of ∼0.17 as a result of the intentionally used ratio of 5–1 of the model compound and 2-EHA (Fig. 1b). For such ratios, pseudo-first order conditions can be expected.

Interestingly, in the presence of Lewis and Brønsted acids, the reaction rate is increased tremendously. Only 1 mol% $p$TsOH decreased the time to reach equilibrium to <10 min. The Lewis acid DBTL catalyses the reaction less efficiently, even when at higher concentrations. Remarkably, also sulphuric acid was found to be much less efficient under similar acidic proton concentrations. This can be rationalized by the tendency of the poorly soluble inorganic anions to form clusters or complexes with the ammonium ions. Thus, acidic protons actually become less available for these reactions, and the addition of these acids do not increase the effective equilibrium concentration of the protonated reaction intermediates. Finally, a very slow exchange reaction is observed at 100 °C when 1% of the cyclic guanidine TBD is added, although TBD is known as an effective organocatalyst for transacylation reactions[26]. Rather than acting as a dual base/H-bond donor and nucleophilic catalyst that could enhance transacylation reactions, it acts solely as a proton

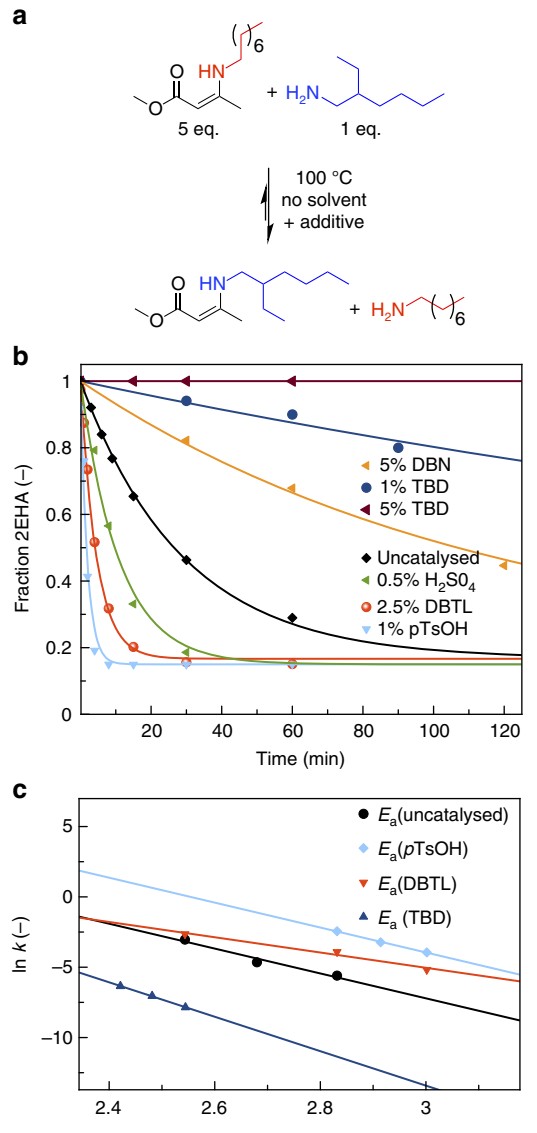

**b**

**c**

**Figure 1 | Kinetic study of VU amine exchange.** (**a**) Used model reaction. (**b**) Decrease of 2-ethylhexyl amine as a function of time at 100 °C in the presence of different additives, which reaches equilibrium for a fraction of remaining 2-EHA of ∼0.17. The amount of catalyst was calculated as mol% versus 2-ethyl hexyl amine (Table 2). (**c**) Arrhenius plot of an extended kinetic study (Supplementary Fig. 2).

scavenger disabling essential proton transfers. This hypothesis is confirmed as an increased concentration of TBD (5%) completely stopped the exchange reaction, while an accelerating effect should be observed in case of any catalytic activity. In addition, also another base such as 1,5-Diazabicyclo[4.3.0]non-5-ene (DBN) slowed down the reaction albeit to a lesser extent. These model study results already indicate that the kinetics of amine exchange of VUs can be readily controlled, using only small amounts of acids and bases.

The most promising additives $p$TsOH, DBTL and TBD were investigated at different temperatures (Supplementary Fig. 2), allowing for the construction of Arrhenius plots and calculation of the activation energies (Fig. 1c; Table 1). The activation energy of the uncatalysed and $p$TsOH-catalysed reactions were within experimental error (∼74 kJ mol$^{-1}$), while a markedly different temperature-dependence was observed for the DBTL-catalysed

**Table 1 | Overview of the activation energies from the model compound study and stress–relaxation experiments of the vitrimer elastomers.**

| Catalyst | $E_a$ model reactions (kJ mol$^{-1}$) | $E_a$ stress–relaxation (kJ mol$^{-1}$) |
|---|---|---|
| Uncatalysed | 73 ± 12 | 81 ± 3 |
| $p$TsOH | 73 ± 3 | 70 ± 4 |
| DBTL | 45 ± 7 | 31 ± 10 |
| TBD | 102 ± 3 | 122 ± 19 |

reactions, giving a much lower activation energy of only 45 (± 7) kJ mol$^{-1}$. From a mechanistic viewpoint, the only small difference in activation energy between the non-catalysed and acid-catalysed samples indicates that the same reaction pathway is followed in both processes and that the faster reaction rates in the presence of a protic acid is the consequence of the higher concentration of active protonated species, as compared to the reference material (Fig. 2a). This explains the markedly faster exchange without a different temperature dependence.

On the other hand, the DBTL-loaded samples showed a strong decrease in activation energy. These observations point to a different reaction mechanism, such as carbonyl activation of the vinylogous urethane by Lewis acid complexation (Fig. 2b). Because of the decreased slope in the Arrhenius plot, exchange reactions could become more significant at lower temperatures with this catalyst. Conversely, the samples inhibited with TBD showed an increased slope and a significantly elevated activation energy to 102 (± 3) kJ mol$^{-1}$, again pointing to a different mechanism, involving the expected addition/ elimination reactions without cationic protonated intermediates (through zwitterionic intermediates), as shown in Fig. 2c.

**Impact of additives on vitrimer materials.** Having established the effects of various additives on model reactions involving low-MW compounds, the influence of additives on the viscoelastic behaviour on vinylogous urethane based polymer networks was examined. Therefore, low $T_g$ vinylogous urethane networks were prepared by mixing priamine 1074 **1**, tris(2-aminoethyl)amine **2**, acetoacetylated pripol **3** and one of the selected additives from the initial screening study. The spontaneous condensation reaction between acetoacetates and amines (Fig. 3) was performed using a small excess of amines versus acetoacetates (a ratio of 0.95) in order to ensure that free amines, required for the exchange reaction, would be available throughout the polymer network. After curing for 6 h at 90 °C, full conversion of the acetoacetates to the vinylogous urethanes was observed according to infrared spectroscopy (Supplementary Fig. 3). Moreover, these curing conditions appeared also sufficient to remove all the water released during the condensation reaction as no more mass loss was observed by thermogravimetric analysis (TGA) after 4 h (Supplementary Fig. 4). Using a monomer ratio of 0.40:0.40:0.95 of **1:2:3**, a network with a $T_g$ of −25 °C, a Young modulus of 2.0 MPa, an elongation at break of 140% and a yield stress of 1.2 MPa was obtained. The mechanical properties of the vitrimers prepared according to this procedure can be tuned by simply changing the monomer ratios (Supplementary Table 2; Supplementary Figs 5 and 6).

The obtained networks were subjected to stress–relaxation experiments in a rheometer (plate–plate geometry), wherein a deformation of 5% was applied and the decrease of stress was measured over time. Since the stress–relaxation behaviour of vitrimers can be described by the Maxwell law $G(t)/G_0 =$

**Figure 2 | Proposed mechanism in the presence of different additives.** (**a**) in neutral and acidic conditions via the formation of an iminium intermediate, (**b**) in the presence of a Lewis Acid, which activates the carbonyl via coordination and stabilizes the zwitter-ionic intermediate, (**c**) in basic conditions via a direct conjugated addition and an unstabilised zwitter-ionic intermediate.

**Figure 3 | Vinylogous urethane vitrimer elastomers.** The monomers (priamine 1, tris(2-aminoethyl)amine 2 and acetoacetylated pripol 3) were used for the synthesis of elastomeric vitrimers together with an additive (Table 3).

exp $(-t/\tau)$, relaxation times were taken when the normalized stress decreased to a value of 0.37 (1/e). The uncatalysed reference network showed a relaxation time of ~10 min at 120 °C. Addition of 0.5 mol% pTsOH versus the total quantity of initial amines in the used monomers, which corresponds to a protonation of ~10 mol% of the excess amines in the resulting networks, shortened the relaxation time to only 2 min. As in the low-MW model experiments, the influence of 0.25% of $H_2SO_4$ and 1.90% DBTL was significant but less pronounced (Fig. 4a). The results obtained for the stress relaxation experiments are in large qualitative and even quantitative agreement with the model compound study since almost a perfect match is observed between the relaxation times and reaction half-lives (Supplementary Fig. 7). Obviously, the remark can be made that the catalyst loadings of model compounds and materials are not identical and the correlations could be a coincidence. Nevertheless, the observed timescales are in good agreement.

When the relaxation times at different temperatures are examined in an Arrhenius plot, a linear relationship can be observed, which is characteristic for vitrimer materials[7]. The samples containing 0.5% pTsOH exhibit a clear downward shift compared to the uncatalysed samples and again only a small distinction between the activation energy can be observed. They were measured as $(81 \pm 3)$ for the uncatalysed and

$(70 \pm 4)$ kJ mol$^{-1}$ for the sample with 0.5% pTsOH from the slopes of the curves in Fig. 4b. These values are comparable to those measured for the model compound exchange reactions $(73 \pm 11$ kJ mol$^{-1})$ and slightly higher than those for the high $T_g$ vinylogous urethane networks we reported previously $(60 \pm 5$ kJ mol$^{-1})$[27]. This rise in activation energy can be ascribed by the more hydrophobic matrix, wherein long aliphatic chains of priamine **1** and acetoacetylated pripol **3** selectively destabilize the cationic reaction intermediates and transition states.

As for the model study with low-MW compounds, these results indicate that the same reaction pathway is followed in both the catalysed and 'uncatalysed' networks, implicating protonated species as crucial reaction intermediates. The faster reaction rates in the presence of a protic acid is thus the consequence of the higher concentration of active protonated species, as compared to the reference material. On the other hand, the DBTL-loaded samples show a strong decrease in activation energy to only 30 ($\pm$4) kJ mol$^{-1}$. These observations are again in line with a very different reaction mechanism, such as carbonyl activation of the vinylogous urethane by Lewis acid complexation wherein the concentration of protonated species is inconsequential. Remarkably, the DBTL catalyst also acts as an inhibitor of the 'protic' pathway, as above a certain temperature the relaxation

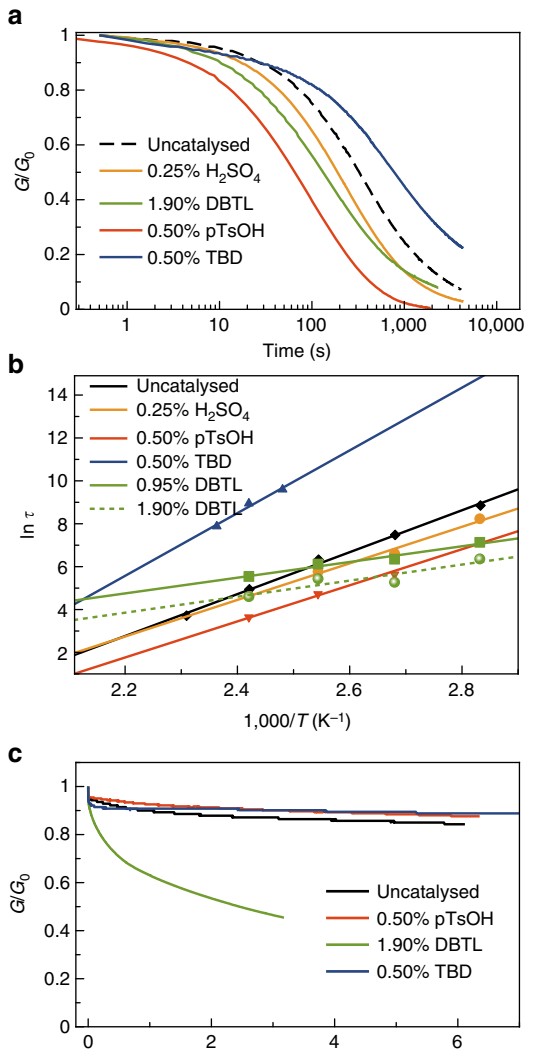

**Figure 4 | Rheology of the VU vitrimer elastomers.** (**a**) Stress–relaxation experiment for networks with acid and base additives at 120 °C and a deformation of $\gamma = 5\%$; (**b**) Arrhenius plot for samples loaded with catalysts. The amount of catalyst was calculated as mol% versus the amine functionalities in the initial monomer mixture; (**c**) Stress–relaxation experiments at 30 °C and a deformation of $\gamma = 5\%$.

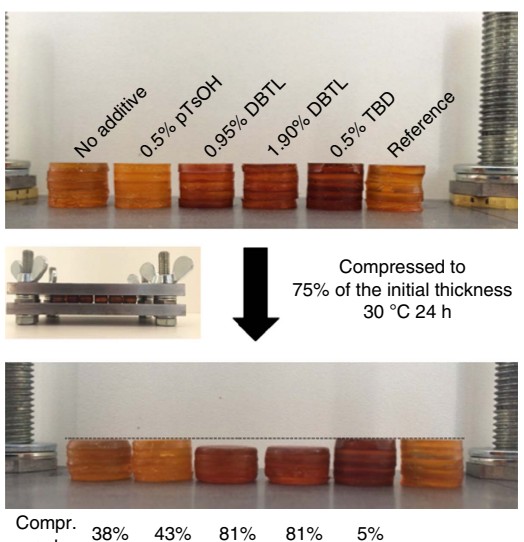

Compr. set: 38% 43% 81% 81% 5%

**Figure 5 | Compression set experiment.** Samples loaded with base (0.5% TBD) showed a great compression set resistance with only 5% permanent deformation. On the other hand, DBTL-loaded samples show a permanent deformation of 85%. Samples without and with 0.5% pTsOH showed a medium compression set resistance with a value of 38% and 43%, respectively. The reference sample was not compressed and served for visual comparison.

becomes slower than in the uncatalysed version. Carboxylate anions can indeed act as proton scavenger for ammonium species in non-aqueous media, giving lower concentrations of alkyl ammonium species.

Because of the decreased slope in the Arrhenius plot, DBTL-catalysed exchange reactions remain more significant at lower temperatures than with other catalysts (lower vitrimer temperature, $T_v$). Conversely, the samples inhibited with the guanidine base TBD showed an increased slope and a significantly increased activation energy of $(122 \pm 19)$ kJ mol$^{-1}$. This difference points again pointing to a very different mechanism not involving addition/elimination reactions to protonated intermediates, but rather going through a direct addition pathway of a neutral amine to a neutral VU (Michael addition). This zwitterionic pathway is much slower but can become fast at higher temperatures.

To investigate the possibility of exchange reactions at room temperature and thus the resistance to creep in these elastomers, further stress–relaxation experiments (small deformation, $\gamma = 5\%$) and compression set experiments (large deformation, 25% compression) were conducted at 30 °C. In the stress–relaxation tests, the uncatalysed reference sample, together with the 0.50% pTsOH and 0.50% TBD-catalysed sample, relaxed $\sim 10\%$ of the initial stress after 6 h (Fig. 4c). Such partial relaxation is not uncommon, even for classical elastomers[28], in particular when considering the intentionally installed network defects in these materials. On the other hand, vitrimer elastomers loaded with DBTL exhibited a strong stress–relaxation, indicating significant network rearrangements at room temperature. Indeed, as anticipated, due to the low-activation energy, the reaction is not sufficiently decelerated at 30 °C in order to effectively freeze the network topology, also indicated by its $T_v$ of $-63$ °C (for all $T_v$ values, see Supplementary Table 3). In the compression set experiments, samples were compressed to 75% of their initial thickness for 24 h at 30 °C. After removal of the applied deformation and recovery time of 30 min, samples were measured again and compared to the initial thickness. The samples without additive and with 0.5% pTsOH had a medium compression set resistance with a permanent deformation of 38 and 43% respectively (Fig. 5). In agreement with the stress–relaxation experiments, the samples with DBTL showed a larger permanent deformation of 81%, indicating a very extensive stress–relaxation. Interestingly, TBD-loaded samples showed excellent resistance towards compression as they almost completely returned to their initial position and performed like a true elastomer (only 5% permanent deformation).

Finally, to demonstrate that the catalytic control of VU vitrimers is not limited to elastomeric networks, rigid VU networks containing 0.5% pTsOH were prepared ($T_g = 87$ °C, $E' = 2.4$ GPa at 30 °C). Stress–relaxation experiments were conducted in torsion geometry[29] with a deformation of 1% and compared to the uncatalysed samples. Similar to the elastomeric networks, a much faster stress–relaxation can be observed, again without a significant change in activation energy (Supplementary Fig. 8). Interestingly, relaxation times as short as 18 s at

150 °C were measured for the acid-catalysed samples. Such fast stress–relaxation enables processing similar to vitreous silica. Indeed, fusilli-shaped samples could be prepared from a flat bar in ~1 min, using a heat gun and a couple of clamps (Supplementary Movie 1).

In summary, we showed that the amine exchange of vinylogous urethanes can easily be controlled using acid and base additives on both low-MW compounds and polymer networks. In this context, three regimes are distinguishable. In neutral or acidic conditions, the amine exchange progresses via an iminium intermediate. When proton exchanges are restricted through the addition of a strong base, direct Michael addition is proposed and finally also a pathway involving direct carbonyl activation via Lewis acids is demonstrated through addition of DBTL. A close correlation between model reactions and mechanical relaxations was observed, confirming that model experiments enable the rational design of vitrimers with predictable and tunable viscoelastic behaviour[7]. These findings significantly expand the scope of vinylogous urethane based vitrimers and introduce a simple chemical strategy to tune the visco-elastic behaviour of vitrimeric elastomers.

## Methods

**Instrumentation.** Nuclear magnetic resonance spectra were recorded on a Bruker Avance operating at 300 MHz. ATR-FTIR spectra were taken on a Perkin-Elmer Spectrum1000 FTIR infrared spectrometer with a diamond ATR probe. A Mettler Toledo TGA/SDTA851 under nitrogen atmosphere or air was used for the thermogravimetric analyses operating at 10 °C min$^{-1}$. Differential scanning calorimetry (DSC) thermograms were recorded on a Mettler Toledo 1/700 system under nitrogen at a heating rate of of 10 °C min$^{-1}$. Dynamic mechanical analysis (DMA) was measured on a SDTA861e DMA from Mettler Toledo. For elastomeric vitrimer samples, stress–relaxation experiments were recorded using a Anton-Paar physica MRC rheometer with a plate–plate geometry of 25 mm using a strain of 5%. Rheology-experiments of the hard samples were performed on a Ares G2 rheometer from TA-instruments in torsion geometry with samples of dimension $(1.3 \times 14.5 \times 22)$ mm$^3$ using an axial force of $-0.01$ N and a deformation of 1%. GC was performed on an Agilent 7890A system equipped with a VWR Carrier-160 hydrogen generator and an Agilent HP-5 column of 30 m length and 0.320 mm diameter. A FID detector was used and the inlet was set to 250 °C with a split injection of ratio 25:1. Hydrogen was used as carrier gas at a flow rate of 2 ml min$^{-1}$. The oven temperature was increased with 20 °C min$^{-1}$ from 50 to 120 °C, followed by a ramp of 50 °C min$^{-1}$ to 300 °C.

**Materials.** Pripol 2033 and priamine 1074 were kindly provided by Croda.

**Synthesis model compounds.** Methyl acetoacetate (1.0 eq) and 2-ethyl hexylamine, octylamine or benzylamine (1.1 eq) were mixed in bulk and heated for 2 h at 90 °C while purging with N$_2$ to remove H$_2$O. The excess of amine was removed for the octyl VU by passing the mixture over a short silica column using ethyl acetate as an eluent. The 2-EH VU or benzyl VU were not purified further as they were only used as a reference.

Yield methyl-3-(octylamino)but-2-enoate: 99%. $^1$H NMR (300 MHz, CDCl$_3$): δ (p.p.m.) = 0.88 (t, 3H, $J = 7$), 1.26–1.38 (m, 10H), 1.51–1.59 (m, 2H), 1.91 (s, 3H), 3.16–3.22 (m, 2H), 3.55 (s, 3H), 4.36 (s, 1H), 8.47 (br s, 1H) (Supplementary Fig. 9). HR-MS(ESI): calculated for C$_{13}$H$_{26}$NO$_2^+$ [M + H]$^+$ 228.1958; found 228.1966.

Yield methyl-3-((2-ethylhexyl)amino)but-2-enoate: 99% $^1$H NMR (300 MHz, CDCl$_3$): δ (p.p.m.) = 0.88 (t, 6H, $J = 7.3$ Hz), 1.27–1.30 (m, 9H), 1.90 (s, 3H), 3.04–3.16 (m, 2H), 3.61 (s, 3H), 4.41 (s, 1H), 8.61 (br s, 1H) (Supplementary Fig. 10). HR-MS(ESI): calculated for C$_{13}$H$_{26}$NO$_2^+$ [M + H]$^+$ 228.1958; found 228.1968.

**Acetoacetylation of pripol2033.** Pripol 2033 (10.0 g, 1 equiv.) and tert-butyla-cetoacetate (6.71 g, 2.3 equiv.) were dissolved in 8.5 ml xylene and 5.5 ml hexane. The mixture was heated to 135 °C in a distillate set-up until the temperature of the vapour dropped below 63 °C. Then the heat was turned up to 150 °C until no more solvent was transferred into the receiving flask. The remainder of the solvent was removed under high vacuum at 80 °C raised until 100 °C, yielding the desired product. No further purification was required.

Yield: 98%. $^1$H NMR (300 MHz, CDCl$_3$): δ(p.p.m.) = 0.79 (m, 6H,), 1.10–1.5 (band, CH2 and CH), 1.57–1.68 (band,), 2.29 (s, 6H), 3.46 (s, 4H), 4.06 (t, 6.75 Hz, 4H). (Supplementary Fig. 11).

**Table 2 | Used equivalents of catalyst for the model compound study.**

| Catalyst | Equiv. |
|---|---|
| No additive | / |
| p-Toluene sulphonic acid | 0.01 |
| 1,5,7-Triazabicyclo[4.4.0]dec-5-ene (TBD) | 0.01 or 0.05 |
| 1,5-Diazabicyclo[4.3.0]non-5-ene (DBN) | 0.05 |
| Dibutyl tin dilaureate | 0.025 |
| Sulphuric acid | 0.005 |

**Table 3 | Used equivalents and amounts for rheological measurements.**

| Reagents | Eq | N (mol) | MW (g mol$^{-1}$) | m (g) |
|---|---|---|---|---|
| Pripol AA | 0.95 | 0.0211 | 710.00 | 15.00 |
| TREN | 0.40 | 0.0089 | 146.23 | 1.30 |
| Priamine | 0.40 | 0.0089 | 547.00 | 4.86 |

**Model compound study.** The N-octyl vinylogous urethane (5 equiv.) was mixed together with 2-ethyl hexylamine (1 equiv.), dodecane (Internal standard, 0.5 equiv.) and the catalyst (Table 2) in a test tube. The resulting mixture was heated at 100 °C. At specified time intervals, samples (~10 mg) were taken and immediately diluted in dichloromethane. The ratio of 2-ethylhexyl amine (3.41 min) and octyl amine (3.67 min) was analysed using GC-FID (Supplementary Figs 1 and 2).

**Kinetic study.** 2-ethylhexyl vinylogous urethane (0.100 g, 1 equiv.) was mixed with benzylamine (0.236 g, 5 equiv.) and a catalyst (0.05 equiv.) in a pressure tube. Sampling was performed by taking a sample with a glass pipette and immediately dissolved in 0.6 ml of benzene-d6 to prevent further conversion. $^1$H- NMR of the samples were taken as soon as possible and the conversion was calculated using the signals at p.p.m. 2.70 of the 2-EHVU (app. t, 2H, -NHCH$_2$CH-) and at 3.72 p.p.m. of the benzyl VU (d, 2H, ArCH2NH-). This procedure was repeated at three different temperatures and k-values were obtained by fitting (Supplementary Note 1; Supplementary Table 1).

**Synthesis of soft vitrimer networks.** Priamine 1074, TREN and acetoacetylated pripol were weighed in this given sequence in a vial (for stoichiometry, see Table 3). The resulting biphasic system was then manually mixed until a homogeneous mixture was obtained. The mixture was poured onto a teflon sheet and manually spread to thickness of around 1 mm and heated for 6 h at 90 °C. The obtained films were compression moulded for 30 min at 150 °C to get fully homogeneous, defect-free samples with a dimension of $\sim (4.5 \times 7 \times 1.7)$ cm$^3$.

For the catalysts, equivalents were calculated as mol% versus the amines of the monomer mixture: pTsOH (0.5 mol%, 0.18 m%): 0.038 g, H$_2$SO$_4$ (0.25 mol%, 0.05 m%): 0.011 g DBTL (0.95 mol%, 1.2 m%): 0.267 g, DBTL (1.9 mol%, 2.4 m%): 0.534 g

TBD was incorporated via swelling. The samples were swollen during 15 min in DCM in which 14.6 mg ml$^{-1}$ TBD was dissolved. The solvent was removed in vacuo overnight.

**Synthesis of hard vitrimer networks.** m-Xylylene diamine (2.111 g, 15.5 mmol), tris(2-aminoethyl)amine (1.774 g, 12.1 mmol) and para-toluene sulphonic acid (0.5 mol%) were weighed in a vial, heated in an oil bath at 80 °C and 1,4-cyclo-hexanedimethanol bisacetoacetate (10,000 g, 32.0 mmol) was added while stirring manually. When a homogeneous liquid mixture was obtained, the mixture was removed from the oil bath and turned gradually white due to phase separation with the water condensate. The resulting white paste was spread out to a film of around 1.3 mm between two teflon sheets using a pre-heated press at 90 °C. After 30 min, the film was cured further in a convection oven for 24 h at 90 °C followed by a post-cure of 30 min at 150 °C.

**Compression set experiments.** Compression set experiments were conducted according to ASTM D395. Disk shape samples with a diameter of 12 mm were cut out and five of these samples were stacked together. These samples were all compressed to exactly the same thickness of 9.21 and heated for 2 h at 140 °C, resulting in samples with exactly the same height. Next, the samples were

compressed to 6.9 mm using a spacer (that is, 76% of its initial thickness) and are put into an air-circulated oven for 24 h at 30 °C. After this period, the specimens are removed from the fixture and after 30 min, their heights are measured. Compression set was calculated using the formula:

$$\text{Compression Set} = \left\{ \frac{(\text{orig. thickness} - \text{final thickness})}{(\text{orig. thickness} - \text{spacer thickness})} \right\} \times 100\% .$$

**Data availability.** The authors declare that the data supporting the findings of this study are available within the paper and its Supplementary Information files.

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

## Acknowledgements

Bernhard De Meyer and Bastiaan Dhanis are acknowledged for laboratory support; Veerle Boterberg, Annemie Houben and Rémi Fournier for the aid on rheology experiments, Croda for providing samples. W.D. thanks the Agency for Innovation by Science and Technology in Flanders (IWT) for a Ph.D. scholarship. F.D.P acknowledges UGent funding (BOF-GOA) and the Program on Interuniversity Attraction Poles initiated by the Belgian State (P7/05).

## Author contributions

The manuscript was written through contributions of all authors. W.D. and M.D. performed all experiments. All authors have given approval to the final version of the manuscript.

## Additional information

**Competing interests:** The authors declare no competing financial interests.

**Publisher's note**: 

