## [Peer Review File · Nature Communications]

Reviewers' comments:

Reviewer #1 (Remarks to the Author):

This manuscript deals with an interesting topic, but y overall feeling is that publication in a high-profile journal is at this stage premature. The main reason for this overall conclusion is that on too many occasions, the authors are quite vague in their explanation of experimental observations. The explanations are typically very generic or hypothetical when it comes to the role of the various additives on the kinetics and mechanism of the reactions. For example, they don't explain why the catalysis by pTsoH is so much better than that by H₂SO₄. The mechanism of the reaction in the presence of TBD is explained as "possibly involving ... zwitterionic intermediates". One suggestion would be to elaborate on the model experiments in order to obtain data that would allow genuine model discrimination to distinguish among different mechanistic pathways. In its present form, the manuscript solely proves that the observations in the vitrimers are qualitatively similar to what is observed in the model experiments. Although perfectly fine for specialised journals in the field of polymer chemistry, a more rigorous study with genuine discrimination among mechanistic pathways would be required for publication in a high-profile journal.

Reviewer #2 (Remarks to the Author):

Various catalysts were used as additives in vinylogous urethane vitrimers in effort to modify the transamidation kinetic profile within the material. This allowed for the viscoelastic properties to be modified. Model studies were performed in order to predict the anticipated behavior of the vitrimers which were synthesized.

Opinion:

This work is of value due to the proven ability to readily tune the rate of transamidation within vinylogous urethane vitrimers. However, the initial appealing aspect of vinylogous urethanes being used for vitrimers is the advantage of being catalyst free. Addition of catalysts, or additives, to this system decreases the appeal, yet also allows one to tune the viscoelastic performance using impressively low levels of catalyst concentration which is advantageous. The work presented is publishable after the following suggested revisions.

Specific Questions/Issues:

1. It is surprising that there is no mention of Leibler's ACS Macro Letters paper "Catalytic Control of the Vitrimer Glass Transition" (ACS Macro Lett. 2012, 1, 789–792) in which they studied the effect of different transesterification catalysts (additives) on tuning the vitrimer's transesterification reaction rate, which is essentially the focus of this study.
2. It appears the model study shows a preference for forming the VU species of 2EHA rather than that of n-octylamine. Is there a reason that a nondegenerate reaction was chosen for the model study as opposed to one whose equilibrium constant would be closer to 1?
3. The materials that are synthesized using TREN contain a tertiary amine in the network that is similar to triethylamine. Perhaps it would be a good idea to study the effect of triethylamine on transamidation of your model compounds. This would be important insight due to the presence of TREN, thus tertiary amine bases, being present in the synthesized networks.
4. P5, L106 – Regarding the mechanism in Figure 1a showing deprotonation of the ammonium species by the VU species, it would be best to show protonation occurring at the carbonyl oxygen of the VU due to the formation of a resonance stabilized cation. Subsequent steps should be adjusted that include a protonated carbonyl oxygen that allows for proton exchange.
5. There needs to be a better discussion that outlines the observed differences of base- and acid-catalyzed transamidation. In the case of the addition of TBD, it is interesting that transamidation becomes slower than the noncatalyzed reaction. This discussion of the effect of TBD is an important focus of the data presented due to its large difference in controlling the viscoelastic properties. The section that covers this discussion should be revised (P7, L156) to sound more coherent. Indeed, I agree that a different type of mechanism occurs. I would advise looking into the literature, starting with J. Org. Chem., 2009, 74, 9490–9496 to aid in discussing the role of

TBD versus other additives.

Small Errors:

1. P2, L30 – place citations after comma
2. P2, L41 – remove space before citation 7
3. P3, L51 – group together the citations to read as “9-23”
4. P3, L58 – insert “(Tg)” after glass transition temperature
5. P4, L77 – remove “transamination” insert “transamidation”
6. P4, L78 – change “functions” to “functionalities”
7. P4, L85 – remove “,”
8. P4, L91 – remove “during polyurethane synthesis”
9. P4, L96 – “GC” and “FID” were not defined
10. P4, L101 – remove “when using” insert “at”
11. P4, L101 – remove “also H2SO4” insert “sulfuric acid”
12. P5, L102 – remove “at the exact same” insert “under similar”
13. P5, L103 – remove “amidine”, insert “cyclic guanidine”
14. P5, L104 – remove “transamination” insert “transamidation”
15. P5, L106 – Figure 1a) – the “+” that is placed before RNH3 should be placed centered to the two substrates in order to prevent confusion of multiple positive charges
16. P5, L111 – italicize T, make g subscript for Tg
17. P6, L123 – remove commas
18. P6, L124 – capitalize t in table and f in figure
19. P6, L131 – remove parentheses before “G(t)”
20. P7, L142 – reference 7 should be moved to after the period
21. P8, L166 – remove “functions” insert “functionalities”
22. P8, L170 – reference 25 should be moved to after the comma
23. P9, L177 – insert “s” after “minute”
24. P10, L205 – reference 7 should be moved to after the period
25. P10, L206 – remove “transamination” insert “transamidation”
26. P11, L217 – italicize T in Tg
27. P11, L220 – insert the unit of distance
28. P12, L252 – GC methods should be explained under the instrumentation section.
29. P13, L258 – the equivalents of TBD found in Table 1 does not match the equivalents shown in Figure 1
30. P13, L267 – Table 2 is not needed.
31. P14, L272 – it is unclear what the percentages in parentheses represent.
32. Proton NMR shifts are reported, yet the spectra are not attached in the supporting information.
33. Supporting information figure and table captions are in calibri

Reviewer #3 (Remarks to the Author):

The manuscript by Denissen, et al. concerns the study of catalyzed stress relaxation of vinylogous urethane vitrimers. The article itself is well written and the study certainly worthy of publication in the journal at hand. However, I believe that there are some discrepancies within the manuscript that deserve careful consideration before publication. In addition, I believe there are some other experiments that need to be performed. Below I have provided a list of concerns:

Main Text

Page 2:

Were the model compound studies run under inert gas? If not, one might expect to see oxidation or amines and/or alkenes in the matrix. Furthermore, TBD would quickly be deactivated in the

presence of carbon dioxide and water. If the experiment was run under inert gas, please clarify in the experimental section. If not, were other side products present in the GC traces? I believe that putting the traces in the SI would be beneficial to the viewership.

Page 5:

Figure 1B: The black data in the figure is listed as "Blanco". I am assuming this was the uncatalyzed model reaction and should be listed as such to be consistent with the other figures.

Page 6:

Line 121: You state that you used a ratio of 0.95:0.40:0.40 of 1:2:3. I think this must be an error and you meant for it to be 0.40:0.40:0.95 of 1:2:3.

Lines 137-140; Figure S4: Please list the full experimental details in Figure S4; for instance, catalyst mol% and temperature. I believe the mol% of the catalyst (2.5 vs. 0.95 or 1.9 mol%) and the ratio of amines to VUs (19:1 and 5:1) is very different between the model compound studies and the materials. Could you please address how this might affect the results if at all? As the samples containing DBTL exhibit the best catalytic properties, it would be a good idea to compare the model compounds for these.

Page 7:

Lines 144-145: Based on Figure 3B, I believe the lower activation energy is attributed to the pTSOH vitrimers. Could you please specify that in this sentence for clarity? Also, what is the error associated with the calculation of the activation energies? Is the difference statistically insignificant between pTSOH and no catalyst as you imply?

Lines 150-151: Is the decrease by 30 kJ/mol or to 30 kJ/mol? I think it would be best to clearly state the actual activation energy here for DBTL for clarity; also please include the errors associated with all the activation energy calculations.

Page 10

Line 200: Movie S1 does not indicate what sample is being used nor does it give an approximate temperature range that the heat gun is applying on the sample. Could you please include this relevant information in the video?

Other concerns and additional experiments:

1) What is the activation energy of the model compound studies? In your original study, this value is in good agreement with uncatalyzed samples. It would be a good idea to determine these for the catalyzed materials to ensure this agreement continues. Although it matches at the one temperature you have performed, it is possible that this could vary once more temperatures are included.

2) Samples produced are tensile tested but the tensile properties are not determined for welded samples. Stress relaxation could be fast due to some catalyzed decomposition of the networks, especially in the presence of catalyst. In order to show the utility of these catalysts, the mechanical properties need to be determined after recycling.

3) What is the freezing transition temperature, T_v , for these samples? You have all of the data necessary to calculate it (modulus and Arrhenius curves), so it should be included for all samples.

Reviewer #4 (Remarks to the Author):

The paper by Denissen et al. provides a simple method to control the covalent exchange kinetics of vinylogous urethane based vitrimers using acid or base additives. Indeed it is desirable if a vitrimer shows little creep at service temperature while maintain a high rate of exchange at

processing temperature. But the work in this paper does not reach this aim. With TBD additives, the creep of vitrimers at 30°C is indeed less than the sample without additives, but the exchange rate at processing temperature (100°C) is slowed down too. With acid additives, the exchange rate at processing temperature is higher, but the creep is more severe than the sample without additives. Even though I like their previous paper (*Adv Funct Mater* 2015, 25, 2451-2457), I am afraid that this paper is not up to the high standards of *Nature Communications*. It is an extension of the AFM paper. But this paper is well-written and the method is good. Given the increasing attention to vitrimers nowadays, maybe this paper can be considered for *Sci.Rep.* or other journals such as *ACS Appl. Mater. Interfaces*.

Dear reviewers,

First of all, we would like to thank the reviewers sincerely for their really valuable suggestions and detailed remarks. Spurred by these comments, we have now performed and included a number of additional experiments, that further strengthen much the content of our manuscript. Significantly, additional small molecule kinetic model experiments now provide a clearer insight into the prevalent reaction mechanisms of the amine exchange reactions and also demonstrate the versatility of the presented approach with regard to the level of control that can be achieved on dynamic material properties.

In the justification text below we provide a point-by-point response to all comments made by the four reviewers.

Reviewer: 1

This manuscript deals with an interesting topic, but my overall feeling is that publication in a high-profile journal is at this stage premature. The main reason for this overall conclusion is that on too many occasions, the authors are quite vague in their explanation of experimental observations. The explanations are typically very generic or hypothetical when it comes to the role of the various additives on the kinetics and mechanism of the reactions. For example, they don't explain why the catalysis by pTsOH is so much better than that by H₂SO₄. The mechanism of the reaction in the presence of TBD is explained as "possibly involving ... zwitterionic intermediates".

One suggestion would be to elaborate on the model experiments in order to obtain data that would allow genuine model discrimination to distinguish among different mechanistic pathways. In its present form, the manuscript solely proves that the observations in the vitrimers are qualitatively similar to what is observed in the model experiments. Although perfectly fine for specialised journals in the field of polymer chemistry, a more rigorous study with genuine discrimination among mechanistic pathways would be required for publication in a high-profile journal.

Fair point. Our discussion of the mechanistic underpinnings of the amine exchange reactions in our original manuscript was indeed vague. Although our explanation of the difference between H₂SO₄ and PTSA remains speculative, we now addressed this issue with a tentative kinetic rationale in text (differential ion pair formation). However, we can now be much more informative on the reaction mechanism of exchange reaction and included quite a number of additional experiments to further strengthen the discussion of the different mechanistic models, including determination of activation energies for different low MW systems.

These data now clearly show three different mechanisms at work: a) protic, b) aprotic, and c) Lewis acid activated pathways. We further show that the protic mechanism is actually rate determining under additive free conditions, and that the system follows a different (higher energy) pathway when a strong base is added to lower the concentration of free ammonium species. Yet another pathway becomes prevalent when a carbonyl-coordinating Lewis acid is added, leading to faster exchange reactions at lower temperatures, but to a different temperature dependence (lower activation energy) that leads to slower exchange at higher temperatures.

In this context, we conducted additional model experiments, which clearly distinguished different mechanistic pathways. Consequently, the section dealing with these experiments is adapted in the revised manuscript (p.5-7). Overall, the predictive value of these mechanistic models now establish a clear way to control viscoelastic properties of vinylogous urea vitrimers, and also offers insight into the factors that can influence amine exchange.

Reviewer: 2

Various catalysts were used as additives in vinylogous urethane vitrimers in effort to modify the transamidation kinetic profile within the material. This allowed for the viscoelastic properties to be modified. Model studies were performed in order to predict the anticipated behavior of the vitrimers which were synthesized.

Opinion:

This work is of value due to the proven ability to readily tune the rate of transamidation within vinylogous urethane vitrimers. However, the initial appealing aspect of vinylogous urethanes being used for vitrimers is the advantage of being catalyst free. Addition of catalysts, or additives, to this system decreases the appeal, yet also allows one to tune the viscoelastic performance using impressively low levels of catalyst concentration which is advantageous. The work presented is publishable after the following suggested revisions.

Specific Questions/Issues:

It is surprising that there is no mention of Leibler's ACS Macro Letters paper "Catalytic Control of the Vitrimer Glass Transition" (ACS Macro Lett. 2012, 1, 789–792) in which they studied the effect of different transesterification catalysts (additives) on tuning the vitrimer's transesterification reaction rate, which is essentially the focus of this study

A line that explicitly refers to this work is added. (p.2-3 line 48-49)

It appears the model study shows a preference for forming the VU species of 2EHA rather than that of n-octylamine. Is there a reason that a nondegenerate reaction was chosen for the model study as opposed to one whose equilibrium constant would be closer to 1?

Actually, there is no preference for the VU species of 2-ethylhexylamine (2-EHA). The final concentration is situated around 0.17 (i.e. 1/6) due to the initial 5:1 ratio used of the octyl-vinylogous urethane and 2-EHA, which was chosen to obtain pseudo-first order conditions. To avoid confusion, a short clarification was added to the manuscript and figure caption, explicitly mentioning this. (p.4 line 95)

The materials that are synthesized using TREN contain a tertiary amine in the network that is similar to triethylamine. Perhaps it would be a good idea to study the effect of triethylamine on transamidation of your model compounds. This would be important insight due to the presence of TREN, thus tertiary amine bases, being present in the synthesized networks.

We agree that the presence of tertiary amines may have a retarding effect on the exchange reaction, as it can scavenge protons similar to TBD. However, since the difference between pKa of the protonated forms of primary amines (10.8) and tertiary amines (11.1)

is actually quite small, we did not anticipate a large influence. Triethylamine may also be a poor substitute for TREN, as internal hydrogen bonds in TREN may be kinetically significant. However, so far we have not observed large qualitative or quantitative differences between TREN-containing and non-TREN containing materials. Consequently, for the additional model studies, we have thus focused on experiments that can distinguish between the mechanistic models.

P5, L106 – Regarding the mechanism in Figure 1a showing deprotonation of the ammonium species by the VU species, it would be best to show protonation occurring at the carbonyl oxygen of the VU due to the formation of a resonance stabilized cation. Subsequent steps should be adjusted that include a protonated carbonyl oxygen that allows for proton exchange.

Actually, in a forward reaction sense, protonation of the enaminone moiety can in theory occur on three different sites: the nitrogen, the central sp^2 -hybridized carbon (as shown in the manuscript) and the carbonyl oxygen. While protonation on the carbonyl oxygen could result in a more stabilised cation (resonance), this might not be the most relevant protonated species along the reaction coordinate. After all, after addition of an amine, this electronic conjugation is lost.

When an amine adds to a conjugated iminium intermediate, it is in fact expected that the final amination-addition product will spontaneously tautomerize to the carbonyl(ester)-isomer shown at the far right of the reaction equations, as enol-forms of esters are much less stable and will simply convert to the ester by a simple intra- or intermolecular proton transfer (N to C). For the actual exchange of amines, the rate determining step will be the backward reaction, *i.e.* the expulsion of the amine from this protonated amination, leading to an iminium intermediate that will quickly tautomerize back to the preferred conjugated and neutral enaminone form.

Thus, the protonated iminium intermediate shown in our original scheme is the most logical intermediate in a backward sense. In order to avoid confusion, we have now included both pathways in the revised scheme, which shows that the relevant intermediates are the enol tautomer of the iminium intermediate or the originally suggested ester-tautomer iminium species. There is some circumstantial evidence for the prevalence of the iminium intermediate, as the ester-imine tautomer can be observed as a minor isomer in some vinylogous ureas.

5. There needs to be a better discussion that outlines the observed differences of base- and acid-catalyzed transamidation. In the case of the addition of TBD, it is interesting that transamidation becomes slower than the noncatalyzed reaction. This discussion of the effect of TBD is an important focus of the data presented due to its large difference in controlling the viscoelastic properties. The section that covers this discussion should be revised (P7, L156) to sound more coherent. Indeed, I agree that a different type of mechanism occurs. I would advise looking into the literature, starting with J. Org. Chem., 2009, 74, 9490–9496 to aid in discussing the role of TBD versus other additives.

See response to comment by reviewer 1 (additional data to clarify difference in mechanisms). With concern to the exact role of TBD, in the recommended JOC article, the authors report that TBD can act as both bifunctional general base/H-bond donor and nucleophile for transacylation reactions. In order to explore this possibility of catalytic activity of TBD additives (in addition to ‘quenching’ the protic pathway), we performed an additional experiment with higher concentrations of the base (5%

TBD versus initial 1%), which actually showed no exchange at all up to 100°C (data added in figure 1b and explained on p. 5 lines 105 - 109). This confirms our hypothesis that TBD acts solely as a base and does not significantly participate in the exchange reaction (as this should lead to a positive concentration dependence, rather than to a negative one).

Small Errors:

All small errors were corrected in the text and ¹H-NMR spectra are added to the SI.

Reviewer: 3

Were the model compound studies run under inert gas? If not, one might expect to see oxidation or amines and/or alkenes in the matrix. Furthermore, TBD would quickly be deactivated in the presence of carbon dioxide and water. If the experiment was run under inert gas, please clarify in the experimental section. If not, were other side products present in the GC traces? I believe that putting the traces in the SI would be beneficial to the viewership.

The model compounds studies were performed under inert gas, which is added in the text (p.4 line 95). No side products were observed during the kinetic experiments. GC-traces are also added in the SI.

Figure 1B: The black data in the figure is listed as "Blanco". I am assuming this was the uncatalyzed model reaction and should be listed as such to be consistent with the other figures.

This inconsistency is corrected.

Page 6 Line 121: You state that you used a ratio of 0.95:0.40:0:40 of 1:2:3. I think this must be an error and you meant for it to be 0.40:0.40:0.95 of 1:2:3.

Indeed, we changed it accordingly to 0.40:0.40:0.95 of 1:2:3.

Lines 137-140; Figure S4: Please list the full experimental details in Figure S4; for instance, catalyst mol% and temperature. I believe the mol% of the catalyst (2.5 vs. 0.95 or 1.9 mol%) and the ratio of amines to VUs (19:1 and 5:1) is very different between the model compound studies and the materials. Could you please address how this might affect the results if at all? As the samples containing DBTL exhibit the best catalytic properties, it would be a good idea to compare the model compounds for these.

The full experimental details are added. As mentioned, matching the conditions of the model compounds and material level perfectly is not possible although we tried to match the experimental set-up of the model compounds as close as possible with those of the material. Furthermore, the chemical and mechanical activation energies (in addition to relaxation times) of low MW compounds are now reported and materials are also in good correspondence. While the similar relaxation times may be

coincidental, the activation energies are more informative and support a similar mechanism in both model and material.

Page 7: Lines 144-145: Based on Figure 3B, I believe the lower activation energy is attributed to the pTSOH vitrimers. Could you please specify that in this sentence for clarity? Also, what is the error associated with the calculation of the activation energies? Is the difference statistically insignificant between pTSOH and no catalyst as you imply?

We have added the individual values with the associated errors for clarification. The activation energies of additive-free and acid-loaded materials are indeed very similar but the difference between the measured values is smaller than the standard deviation on our measurement. However, the activation energies for the model studies for the uncatalysed and PTSA-catalysed reaction actually converge to a statistically indistinguishable number, further confirming the operation of a protic mechanism, even when no acid catalyst is added (relying on the small equilibrium amounts of ammonium and iminium species).

Lines 150-151: Is the decrease by 30 kJ/mol or to 30 kJ/mol? I think it would be best to clearly state the actual activation energy here for DBTL for clarity; also please include the errors associated with all the activation energy calculations.

The activation energy decreased to 30 kJ/mol, which is made more clear in the text. We have also added a Table with all relevant Ea. Finally, also the measuring error is added (p.10 line 195).

Other concerns and additional experiments:

What is the activation energy of the model compound studies? In your original study, this value is in good agreement with uncatalyzed samples. It would be a good idea to determine these for the catalyzed materials to ensure this agreement continues. Although it matches at the one temperature you have performed, it is possible that this could vary once more temperatures are included.

Additional experiments on model compounds are performed and added to the text, together with a more elaborated discussion. (p. 6-7)

Samples produced are tensile tested but the tensile properties are not determined for welded samples. Stress relaxation could be fast due to some catalyzed decomposition of the networks, especially in the presence of catalyst. In order to show the utility of these catalysts, the mechanical properties need to be determined after recycling.

Tensile tests of the uncatalyzed and acid catalyzed samples are added (Figure S6). After the first recycling, a decrease in modulus is observed while no further deterioration of mechanical properties occurs after additional recycling cycles. Furthermore, no different behavior of the uncatalyzed and acid-catalyzed samples were observed, indicating that the faster stress-relaxation of acid catalyzed samples is not caused by catalyzed decomposition.

What is the freezing transition temperature, T_v , for these samples? You have all of the data necessary to calculate it (modulus and Arrhenius curves), so it should be included for all samples.

A table with the calculated T_v -values is added to the supporting info (table S3) and referred to in the text (p. 12 line 224).

Reviewer #4 (Remarks to the Author):

*The paper by Denissen et al. provides a simple method to control the covalent exchange kinetics of vinylogous urethane based vitrimers using acid or base additives. Indeed it is desirable if a vitrimer shows little creep at service temperature while maintain a high rate of exchange at processing temperature. **But the work in this paper does not reach this aim.** With TBD additives, the creep of vitrimers at 30°C is indeed less than the sample without additives, but the exchange rate at processing temperature (100°C) is slowed down too. With acid additives, the exchange rate at processing temperature is higher, but the creep is more severe than the sample without additives. Even though I like their previous paper (*Adv Funct Mater* 2015, 25, 2451-2457), I am afraid that this paper is not up to the high standards of *Nature Communications*. It is an extension of the AFM paper. But this paper is well-written and the method is good. Given the increasing attention to vitrimers nowadays, maybe this paper can be considered for *Sci.Rep.* or other journals such as *ACS Appl. Mater. Interfaces*.*

First, we want to clarify that the aim of our work was not to have vitrimers with no exchange at service temperature and at the same time *superior* exchange kinetics at processing temperatures. The main aim of our work was to explore the mechanism of the exchange reactions (which was never studied before in detail) in order to possibly identify crucial factors that may control the exchange kinetics. In this way, the scope of the chemical platform of vinylogous urethane vitrimers would be greatly expanded, as the only way to control the exchange rates - and thus stress relaxation times - up to now was to either 'freeze' the networks at service temperature by using high Tg materials, or by introducing less or more network defects (free amines), with a different network architecture also unavoidably resulting in a change of network properties.

We have found that very small amounts of simple additives indeed significantly and predictably influence the exchange kinetics, both on model systems and on materials. We have also elucidated three different activation modes for VU exchange, with varying activation energies, giving a 'tunable' and predictable rheology profile for essentially one and the same material. The most significant finding, however, is that the exchange in our original networks is – in fact – already an acid catalysed (or at least proton mediated) reaction. This has then led to the remarkable feature of VU vitrimers that the exchange rate can actually be slowed down by adding very small amounts of a strong base, thus effectively preventing or at least obstructing the formation of ammonium ion intermediates.

As the concept of deliberately slowing down exchange reactions in a vitrimer to a desired level is entirely new to the field, we very strongly believe that this feature is more than just an extension of our original system, and should be of a much wider interest than just the vitrimer research community. The results may also be of direct relevance to other researchers that look at dynamic covalent bond formation, such as

self-healing, dynamic combinatorial chemistry or the preparation of nanoparticles. Indeed, it now allows the design of 'soft' elastomer networks, using exactly the same monomers and stoichiometry, but wherein creep can be precisely controlled by controlling the amount of free acid (ammonium) in the network. For high T_g materials, the exchange can be made very swift.

Reviewers' comments:

Reviewer #1 (Remarks to the Author):

The authors have adequately taken into account my comments on the first submission. The addition of extra experimental work and extension of the discussion regarding the model compounds has improved the manuscript to a level that I find suitable for publication in Nature Communications.

Reviewer #3 (Remarks to the Author):

I am more than satisfied with the revisions that have been performed on the manuscript to my comments as well as the comments of the other reviewers. The authors responded adequately to all concerns presented and performed all additional experiments asked of them.

Reviewer #4 (Remarks to the Author):

If the intention of this paper is to deal with the control of the viscoelastic properties, the authors should make this even clearer. By the end of the second paragraph of this paper, the authors said, "Thus, vitrimer elastomers are very challenging materials as a precise control of the kinetic of exchange is required to enable preferentially fast processing at elevated temperature (high rate of exchange) and dimensional stability, i.e. no creep at service temperature (no or very low rate of exchange)." This may mislead the readers to think that the work in this paper is to enable "preferentially fast processing at elevated temperature (high rate of exchange) and dimensional stability, i.e. no creep at service temperature (no or very low rate of exchange)" by precise control of the kinetic of exchange.

It is true that this work is the first time to investigate the kinetics of vinylogous urethane vitrimers, but it is not the first time that the mechanism of exchange reactions is investigated. In the field of dynamic covalent chemistry, the mechanism of many exchange reactions have been thoroughly discussed using small molecules. In the field of vitrimers, the catalyst effect on the exchange reaction has also been studied in ACS Macro Lett.2012,1,789-792. It is certainly very predictable that the catalyst which can affect the exchange reaction will affect the viscoelasticity of the polymer network.

For different exchange reactions, the catalyst or the method which can promote or slow down the exchange is very different. The method used here is hardly generalized to other kind of vitrimers, for other kind of vitrimers are based on different kind of exchange reactions. The vinylogous urethane vitrimer is a small class of vitrimers, compared to the epoxy vitrimers, polyurethane vitrimers and other kind vitrimers. Therefore, the authors need more convincing justification on why the work here is highly important and suitable for Nat.Commun.

One more problem is that, on one hand, the authors claim that TBD can slow the reaction because it is a base, on the other hand, the authors point out in line 117, that TBD " acts solely as a proton scavenger disabling essential proton transfers", which means that its effect on the reaction is not because it is a base. Moreover, to generally claim that base slows down the reaction, the authors needs more bases to verify such claim, for only one example of TBD is not enough.

We would like to thank the reviewers sincerely for their second set of suggestions and remarks and are delighted about the positive reviews. To meet the additional comments raised by reviewer 4, we now included the results of an additional experiment, which indeed confirm our claim that bases slow down the reaction.

In the justification text below we provide a point-by-point response to the comments made by reviewer 4.

Reviewer #4 (Remarks to the Author):

If the intention of this paper is to deal with the control of the viscoelastic properties, the authors should make this even clearer. By the end of the second paragraph of this paper, the authors said, "Thus, vitrimer elastomers are very challenging materials as a precise control of the kinetic of exchange is required to enable preferentially fast processing at elevated temperature (high rate of exchange) and dimensional stability, i.e. no creep at service temperature (no or very low rate of exchange)." This may mislead the readers to think that the work in this paper is to enable "preferentially fast processing at elevated temperature (high rate of exchange) and dimensional stability, i.e. no creep at service temperature (no or very low rate of exchange)" by precise control of the kinetic of exchange.

We thank the reviewer for this constructive comment and further clarified this point by saying on line 50 :

" Leibler and co-workers introduced the concept and demonstrated that vitrimers macroscopic dynamics can be controlled by changing the concentration or the nature of the epoxy-based transesterification vitrimers can be controlled by changing catalysts.⁸ Still, for transesterification, the exchange reaction remains slow and vitrimer topology freezing transition occurs at relatively high catalyst loadings and temperatures, even for the most efficient metal catalysts"

and on line 63 we included a clarification:

" In this work, we aimed to use simple organic additives that enable the precise control of exchange kinetics and subsequent viscoelastic properties of vitrimers. We show that the exchange reactions can be accelerated or slowed down at will, and such control enables to achieve desired thermo-mechanical properties, e.g. control of compression set in elastomeric vitrimers."

Furthermore, we like to stress that our title also explicitly reveals this purpose.

It is true that this work is the first time to investigate the kinetics of vinylogous urethane vitrimers, but it is not the first time that the mechanism of exchange reactions is investigated. In the field of dynamic covalent chemistry, the mechanism of many exchange reactions have been thoroughly discussed using small molecules. In the field of vitrimers, the catalyst effect on the exchange reaction has also been studied in ACS Macro Lett.2012,1,789-792. It is certainly very predictable that the catalyst which can affect the exchange reaction will affect the viscoelasticity of the polymer network.

This paper (reference 8), which was published by one of the co-authors, indeed is the first paper about catalytic control of vitrimers, more specifically about the first generation, glassy epoxy-based vitrimers. Whereas there, catalyst is added (and has to be added) to allow the exchange reaction by lowering the activation energy, we have found that catalysts in our vinylogous urethane materials can actually slow down exchange reactions, while either highering (base) or lowering (Lewis acid) the activation energy, i.e. the temperature dependence. This gives an unprecedented and also rather unexpected control of the viscosity profile as both the position and the slope of the viscosity can be altered, opening up many more applications for these polymeric materials.

For different exchange reactions, the catalyst or the method which can promote or slow down the exchange is very different. The method used here is hardly generalized to other kind of vitrimers, for other kind of vitrimers are based on different kind of exchange reactions. The vinylogous urethane vitrimer is a small class of vitrimers, compared to the epoxy vitrimers, polyurethane vitrimers and other kind vitrimers. Therefore, the authors need more convincing justification on why the work here is highly important and suitable for Nat.Commun.

We agree that – in a strict chemical sense - vinylogous urethane vitrimers are representing a vitrimer subclass, although it is an important and growing one because of short relaxation times and ease of scalability. The vinylogous exchange reaction can however at least in principle be transposed to a number of resin matrices, such as epoxies, provided that a number of ‘vinylogous’ crosslinks are built in (eg for amine hardened epoxies). We have already started investigations along these lines - using different polymer matrices - and we are sure other groups are exploring the same options. These examples will soon appear in the literature. As some of these topics are subject of our ongoing research, we did not highlight these ideas. Furthermore, we are convinced that the mechanistic insight into the vinylogous exchange and the level of provided ‘rational’ kinetic control will be useful for the wide field of dynamic covalent chemistry research in different areas beyond the field of vitrimers.

One more problem is that, on one hand, the authors claim that TBD can slow the reaction because it is a base, on the other hand, the authors point out in line 117, that TBD “acts solely as a proton scavenger disabling essential proton transfers”, which means that its effect on the reaction is not because it is a base. Moreover, to generally claim that base slows down the reaction, the authors needs more bases to verify such claim, for only one example of TBD is not enough.

Since a (Bronstedt) base scavenges protons by definition, we believe it is appropriate to surmise that the slowing-down effect is due to the basicity/proton scavenging of TBD. To further demonstrate this point, we now also provide an additional example of an added base (the amidine base DBN) to confirm our claim that bases – in general - slow down the reaction. Therefore, we showed that the reaction kinetics are also slowed down using the weaker base 1,5-diazabicyclo(4.3.0)non-5-ene (DBN) that, as expected, shows a similar effect as TBD (Figure 1a and added on line 117). This is perfectly in line with the trends expected from the standard pKaH values for amines, amidines and guanidines (resp 10, 12 and 14).